# Psychological Stress and Salivary Cortisol Levels in Patients with Plaque Psoriasis

**DOI:** 10.3390/jpm11111069

**Published:** 2021-10-23

**Authors:** Paolo Gisondi, Davide Geat, Francesco Bellinato, Laura Spiazzi, Elisa Danese, Martina Montagnana, Giuseppe Lippi, Giampiero Girolomoni

**Affiliations:** 1Section of Dermatology and Venereology, Department of Medicine, University of Verona, 37126 Verona, Italy; davidegeat@outlook.it (D.G.); francesco.bellinato@gmail.com (F.B.); laura.spiazzi@gmail.com (L.S.); giampiero.girolomoni@univr.it (G.G.); 2Section of Clinical Biochemistry, Department of Neuroscience, Biomedicine and Movement, University of Verona, 37129 Verona, Italy; elisa.danese@univr.it (E.D.); martina.montagnana@univr.it (M.M.); giuseppe.lippi@univr.it (G.L.)

**Keywords:** HADS, PSS, psoriasis, psychological stress, salivary cortisol

## Abstract

Psychological stress has long been recognized as a trigger for plaque psoriasis, and preliminary evidence suggests that psoriasis could be associated with alterations in the hypothalamic-pituitary-adrenal (HPA) axis, resulting in impaired cortisol response to stress. This study aimed to investigate psychological stress, anxiety, depression and salivary cortisol in psoriatic patients. A cross sectional study involving 126 adult patients with plaque psoriasis and 116 adult healthy controls was conducted. Demographic, clinical data, Perceived Stress Scale (PSS) and Hospital Anxiety and Depression Scale (HADS) were collected. Cases and controls were asked whether they felt stressed in the last month, whilst psoriatic patients were also interrogated whether they found that psoriasis could have been worsened by stress. Moreover, 54 randomly selected subjects (27 psoriasis patients and 27 controls) underwent salivary cortisol testing at 8 am. PSS, HADS depression and anxiety subscales were significantly higher in psoriatic patients than in controls (17.2 ± 0.6 vs. 15.1 ± 0.8 *p* = 0.0289), (9.5 ± 0.3 vs. 6.2 ± 0.3 *p* < 0.001) and (8.2 ± 0.4 vs. 4.2 ± 0.3 *p* < 0.001), respectively. A higher rate of psoriatic patients reported feeling stress over the last month (45% vs. 19%, *p* < 0.001), and stress was considered a potential trigger for psoriasis flare-ups in 69% of cases. Psoriasis was strongly associated with higher PSS and HADS scores independently of sex, body mass index, diabetes, hypertension, dyslipidemia, and occupational status. Salivary cortisol was significantly lower in psoriatic patients compared to controls (9.6 ± 0.5 vs. 14.0 ± 1.1 nmol/L, *p* < 0.001). In conclusion, psoriasis was associated with higher psychological stress, anxiety and depressive symptoms, and with impaired cortisol response to stress.

## 1. Introduction

Psoriasis is a chronic immune-mediated skin disease affecting 0.51–11.43% of the adult worldwide population [1]. Psychological stress has been for long recognized as a trigger factor in psoriasis, and a systematic review of the literature found that a probable temporal association between psychological stress and onset, recurrence, and severity of psoriasis [2]. The exact mechanisms behind this association have not been fully elucidated, though alterations in hypothalamic-pituitary-adrenal (HPA) axis resulting in an altered response to stress could be demonstrated in psoriasis by serum and salivary cortisol measurements [3,4,5,6,7,8], thus being thought to have pathogenetic implications [9]. Despite the large number of studies on stress in psoriasis, limited data are currently available on salivary cortisol in psoriatic patients. We conducted an observational study aimed at accessing psychological stress and salivary cortisol levels in psoriatic patients.

## 2. Methods

Consecutive patients attending the University Hospital of Verona for psoriasis (cases) or melanoma screening (controls) between 1 January 2019 and 30 May 2019 were eligible for this study. Inclusion criteria for cases were age >18 years, clinically confirmed diagnosis of chronic plaque psoriasis, lack of systemic treatments for psoriasis for at least 6 months. Exclusion criteria were diagnosis of non-plaque psoriasis such as pustular or erythrodermic psoriasis, cognitive dysfunction that might prevent understanding of administered questionnaires, pregnancy, systemic therapies (e.g., synthetic, or biologic disease modifying anti rheumatic drugs, or phototherapy). Inclusion criteria for controls were age >18 years, hospital admission for routine melanoma screening, i.e., non-urgent scheduled dermatological visits for patients without known suspicious cutaneous lesions or previous history of melanoma. Controls with past or current history of psoriasis, any other inflammatory skin disease or cognitive dysfunction were excluded. Demography, body mass index (BMI), smoking habit, level of education (i.e., <high school diploma, high school diploma, university degree), employment status (i.e., employed, unemployed, retired), comorbidities (i.e., psoriatic arthritis, obesity, diabetes, arterial hypertension, hypercholesterolemia), psoriasis duration (in years) and severity according to PASI (Psoriasis Area and Severity Index) were collected. Patients were classified as having psoriatic arthritis (PsA) when fulfilling the CASPAR criteria [10]. Patients were diagnosed with obesity in the case of BMI ≥ 30; diabetes in case of fasting plasma glucose ≥ 126 mg/dL (i.e., ≥7 mmol/L) or history of medical confirmed diagnosis or antidiabetic treatment; arterial hypertension in case of systolic/diastolic blood pressure ≥ 130/85 mmHg or history of medical confirmed diagnosis or receiving anti-hypertensive treatment; hypercholesterolemia in case of total cholesterol ≥ 200 mg/dL (i.e., ≥5.2 mmol/L), LDL cholesterol ≥ 100 mg/dL (i.e., ≥2.6 mmol/L), or antilipemic treatment. Patients with psoriasis and controls voluntarily agreed to participate in the study, and written informed consent was obtained from each study participant. The study was carried out according to the principles expressed in the Declaration of Helsinki.

### 2.1. Stress Evaluation

Patients completed the validated Italian version of Perceived Stress Scale (PSS; range 0–40) [11]. The PSS represents a 10-item questionnaire designed to evaluate the degree to which subjects perceive life situations as stressful [12]. The PSS is not linked to specific events; instead, its items refer to appraisal of events over the previous month that the subject rates on a five-point scale, with higher scores indicating more perceived stress. This scale was selected for use in the present study as it demonstrated good reliability (both internal consistency and test–retest) and good validity [12]. Moreover, the PSS displayed greater accuracy in predicting a number of health outcomes compared with other measures that add up the number of events experienced in a certain period [12]. The three versions of PSS have been evaluated for use with Italian workers, known as PSS-14, PSS-10, and PSS-4, but the psychometric properties of PSS-10 are greater than those of PSS-14 and PSS-4 [13]. The mean PSS score assessed in a large random sample of the American population (*n* = 2387) was 19.62 (standard deviation [SD] 7.49) [14]. In case the PSS total score ranges from 0 to 13, the patient is managing his/her emotions well and is adequately discharging the tensions of daily and working life. In case the total score ranges from 14 to 26, the patient starts having difficulties in coping optimally to load of stress and emotions he/she is experiencing. In case the total score ranges between 27–40, the patient reached a level of stress that compromises seriously his/her physical and mental health.

Furthermore, to evaluate the awareness of being stressed, cases and controls were also asked to give a yes/no answer to the following question: “Did you feel stressed in the last month?” To assess the potential influence of stress in the worsening psoriasis, patients with psoriasis were asked to give a yes/no answer to the question: “Is your psoriasis worsened by stress?”

### 2.2. Anxiety and Depression

The Hospital Anxiety and Depression Scale (HADS; range 0–42 comprised of 0–21 for anxiety and 0–21 for depression) was used to assess anxiety and depression in the study population. The HADS is a patient-reported anxiety (HADS-A) and depression (HADS-D) assessment comprising seven questions for each subscale (anxiety or depression), with responses for each item ranging from scores of 0 to 3 (totaling 0–21) and higher scores indicating more severe symptoms. According to Snaith and Zigmond [15], the scores on both scales can be interpreted in the following manner: 0–7 = normal, 8–10 = mild, 11–14 = moderate and 15–21 = severe. The HAD has good validity and internal reliability [16]. A cut-off of HADS score ≥ 11 was used to identify patients with moderate to severe clinical signs of anxiety and depression.

### 2.3. Salivary Cortisol Measure

Fifty-four age- and sex-matched subjects (27 psoriasis patients and 27 controls) underwent salivary cortisol testing at 8 am. Exclusion criteria for salivary testing were pregnancy, thyroid disease, oral steroidal treatment, practicing intense physical activity or working on night shifts. Subjects were instructed not to brush their teeth, eat, smoke or drink coffee or alcohol up to 8 h before sampling. A braided cotton dental rope was then placed in mouth and left in place for approximately 5 min to collect saliva. Saliva samples were obtained using Sarstedt’s Cortisol-Salivette^®^ (SARSTEDT AG & Co, Nümbrecht, Germany). In the biochemistry laboratory, the samples were centrifuged, refrigerated, and then analysed for cortisol by liquid chromatography with tandem mass spectrometry with the MassChrom^®^ kit (Chromsystems Instruments & Chemicals GmbH, Munich, Germany).

### 2.4. Statistical Analysis

Normally distributed continuous data were presented as mean ± SD, whilst categorical data were shown as percentages and frequencies. Chi squared test and *t*-test were used as appropriate to compare psoriatic patients and controls. Pearson’s correlation was used to assess the relation between salivary cortisol, PSS and PASI in psoriatic patients. Multiple linear regression analysis was also applied to test the association of psoriasis with stress, anxiety, and depression scores. The covariates included in multivariable regression were according to their biological plausibility and/or statistical association with psoriasis in univariate analyses. *p*-values < 0.05 were considered to be statistically significant. Statistical analyses were performed using Stata version 13 (Stata Corp, TX, USA).

## 3. Results

The results of the study are summarized in Table 1. One hundred and twenty-six adult psoriatic patients and 116 controls were recruited. The mean age of psoriatic patients was 53.8 ± 14.2 years, with a mean PASI of 15.8 ± 4.2. The educational and occupational background of psoriatic patients and controls was significantly different (*p* < 0.001), with higher percentage of non-graduates and retired among the former. A significantly higher prevalence of obesity, arterial hypertension, diabetes mellitus and dyslipidemia was also found among psoriatic patients, whilst the rate of diagnosed psychiatric diseases was similar between the two groups (one patient with depression in each group). Furthermore, PSS and HADS scores were significantly higher in psoriatic patients (*p* = 0.03 and *p* < 0.001, respectively). This was confirmed in multivariate regression analysis (Table 2), where psoriasis was strongly associated with higher PSS and HADS scores after adjustment for sex, BMI, diabetes, hypertension, dyslipidemia, unemployed and retired occupational status. The percentage of cases and controls who reported feeling stressed in the previous month was 45.2% (*n* = 57) and 19% (*n* = 22; *p* < 0.001), while 69% (*n* = 87) of psoriasis patients reported stress as a trigger for psoriasis flare-ups. With respect to the subgroups of psoriatic patients and controls who underwent cortisol testing, there were no significant differences in BMI (25.1 ± 4.0 vs. 24.5 ± 4.7 kg/m^2^, *p* = 0.6), incidence of diabetes (*n* = 3 vs. *n* = 1, *p* = 0.6), hypertension (*n* = 9 vs. *n* = 7, *p* = 0.8) or dyslipidemia (*n* = 4 vs. *n* = 3, *p* = 1), and none reported psychiatric disease or treatment with psychiatric medications. Salivary cortisol levels were found to be significantly lower in psoriatic patients than in controls (9.6 ± 0.5 vs. 14.0 ± 1.1 nmol/L, *p* < 0.001). Among psoriatic patients, no association was found between salivary cortisol and PSS (*r* = −0.24, *p* = 0.23) or PASI (*r* = −0.12, *p* = 0.55).

## 4. Discussion

Psychological stress has long been recognized as an important trigger for psoriasis, and this was confirmed by the present study, where up to 69% psoriatic patients reported stress as trigger factor for psoriasis flare-ups. This is in keeping with results of a recent systematic review of the literature, which could identify stress as psoriasis trigger in up to 88% patients [17]. At our evaluation, the elevated PSS scores among psoriatic patients (indicating significantly higher stress in psoriatic patients) also confirmed this association. In the present study, anxiety and depression were also markedly enhanced in psoriatic patients, as highlighted by the significant differences in HADS scores between psoriatic patients and controls. Such a worrisome link between psoriasis, anxiety and depression is also well-known. In a multicenter observational cross-sectional study, depression, and anxiety (evaluated through the Hospital Anxiety and Depression Scale) were reported in as many as 22.7% and 13.8% of psoriatic patients [18]. Furthermore, anxiety and depression were shown to be associated with psoriasis severity [19,20]. Several factors could justify the association between psoriasis and psychological stress, anxiety, and depression. A crucial factor may be stigmatization due to psoriasis, which has been found in a study to involve up to 73% of all patients [21]. This may lead, in turn, to a sense of inferiority, distorted body image, anxiety, depression, and overall reduction of quality of life, involving professional, social and sexual life [22,23]. In a large study as many as 79% of psoriatic patients reported that psoriasis had a negative impact on their lives [23]. In addition, psychological stress and psychiatric disorders can not only result from, but may also contribute to progression of psoriasis, possibly because they share similar pathogenetic mechanisms. For example, pro-inflammatory cytokines were found to be elevated in both psoriasis and depression, suggesting that inflammation may be a driving factor in the progression of both diseases [24]. If, on one hand, high cytokine levels in psoriatic patients may contribute to development of depression, on the other hand depression, due to its proinflammatory state, can worsen psoriasis [25]—thus resulting in creation of a vicious circle. Psychological stress was also shown to cause an increase of proinflammatory cytokines such as interleukin (IL)-6 and IL-1β [26]. However, other pathways (including the hypothalamic–pituitary–adrenal [HPA] axis, sympathetic–adrenal–medullary axis and peripheral nervous system) may also explain the relationships between stress and psoriasis [26]. 

An interesting finding emerged from our study was the presence of significantly lower levels of salivary cortisol in psoriatic patients than in controls, despite higher stress levels of the former. This is in agreement with findings of previous studies which suggested that that psoriatic patients may have an impaired cortisol response to stress [3,4,5,6,7]. Richards et al. [3] found no significant difference in salivary cortisol between psoriatic patients and controls. However, the authors reported lower levels of serum and salivary cortisol following a social performance stressor in patients with stress-responsive psoriasis compared to the nonstress-responsive group [3]. Similar results were also found by Evers et al. [4], who found that psoriatic patients with persistently higher levels of daily stressors had lower mean serum cortisol values than patients with lower levels of daily stressors. Furthermore, Robati et al. [5] reported a significant increase in serum cortisol levels after psoriatic treatment. These studies suggest that some psoriatic patients may have reduced cortisol response. Such inability to produce an adequate upregulation of cortisol in response to stress is thought to carry important pathogenetic implications given the immunosuppressive action of cortisol [8]. Indeed, glucocorticoids inhibit the differentiation of T helper cells into Th1 and increase the production of Th2 cytokines [27,28,29,30]. In psoriatic patients with reduced cortisol production in response to stress, however, this shift towards a Th2 response occurs to a much lower extent, allowing persistently elevated Th1 cytokines to fuel inflammation [9]. However, our understanding of the HPA axis in psoriasis is still far from complete, as conflicting results have been published by Brunoni et al. [6], who found a direct correlation between bedtime salivary cortisol and PASI (with patients with higher psoriasis severity having elevated salivary cortisol levels). Furthermore, mechanisms other than serum cortisol levels alterations may be involved in the pathogenesis of psoriasis, as cutaneous glucocorticoidogenesis and expression of glucocorticoid receptors were also found to be inhibited in psoriatic skin [31,32].

This study was limited by the small population size and the lack of follow-up data. Furthermore, the findings of the present study may not be generalizable to all psoriasis patients, as the vast majority of the study population had moderate-to-severe psoriasis. The choice of patients who come to hospital for routine melanoma screening as a control group is a limitation of the study because such patients could be stressed for the motivation of the visit. However, this bias may have only diluted a difference that has been observed in patients with psoriasis. Lastly, overweight and obesity could act as confounding factors for the assessment of cortisol in psoriasis in that they are associated with reduced cortisol levels [33]; however, this should not represent a bias of this study given that BMI did not differ significantly between the two subgroups of psoriatic patients and controls who underwent cortisol testing.

Moreover, the presence of a control group and the simultaneous evaluation of salivary cortisol and stress (using a standardized scale for the latter) represent important strengths in this clinical investigation.

In conclusion, psoriasis was found to be associated with significantly higher psychological stress, anxiety, and depressive symptoms. Salivary cortisol levels were lower in psoriatic patients compared to controls despite the more elevated psychological stress of the former, supporting the existence of an altered cortisol stress response in psoriasis.

## Figures and Tables

**Table 1 jpm-11-01069-t001:** Clinical and sociodemographic characteristics of patients with psoriasis and controls.

	Psoriasis (*n* = 126)	Controls (*n* = 116)	*p*-Value
Sex (male/female)	82/44	50/66	0.001
Age (years)	53.8 ± 14.1	52.1 ± 6.6	0.227
PASI	15.8 ± 4.2		
Psoriasis duration (years)	21.6 ± 14.4		
Body mass index (kg/m^2^)	26.6 ± 4.4	24.3 ± 3.8	<0.001
Obesity, *n* (%)	25 (19.8)	8 (6.9)	0.003
Arterial hypertension	37 (29.4)	12 (10.3)	<0.001
Diabetes mellitus	12 (9.5)	3 (2.6)	0.025
Dyslipidemia	16 (12.7)	6 (5.2)	0.042
Personal history of psychiatric diseases	1 (0.8)	1 (0.9)	0.953
Use of benzodiazepines	2 (1.6)	2 (1.7)	0.934
Occupational status			<0.001
Employed	90 (71.4)	108 (93.1)	
Unemployed	6 (4.8)	6 (5.2)	
Retired	30 (23.8)	2 (1.7)	
Educational level			<0.001
Primary	22 (19.0)	3 (2.4)	
Lower secondary	47 (40.5)	5 (4.0)	
Upper secondary	46 (39.7)	56 (44.4)	
Degree	11 (9.5)	52 (41.3)	
HADS			
Depression subscale	9.5 ± 0.3	6.2 ± 0.3	<0.001
Anxiety subscale	8.2 ± 0.4	4.2 ± 0.3	<0.001
PSS	17.2 ± 0.6	15.1 ± 0.8	0.0289

PSS, Perceived Stress Scale; HADS, Hospital Anxiety and Depression Scale.

**Table 2 jpm-11-01069-t002:** Multiple linear regression models assessing the associations of Perceived Stress Scale and Hospital Anxiety and Depression Scale with psoriasis.

Variable	β (95% CI)	*p*-Value
PSS *	2.29 (0.09–4.48)	0.04
HADS-A *	3.07 (2.17–3.96)	<0.001
HADS-D *	3.78 (2.96–4.60)	<0.001

Sample size, *n* = 242. Data are expressed as β 95% CI as assessed by univariate (unadjusted) or multivariable linear regression analysis. * Multivariable regression models adjusted by sex, BMI, diabetes, hypertension, dyslipidemia, unemployed and retired occupational status. PSS, Perceived Stress Scale; HADS, Hospital Anxiety and Depression Scale; CI, confidence interval.

## Data Availability

The data presented in this study are available on reasonable request from the corresponding author. The data are not publicly available due to privacy.

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
