# Peer review of "Psychological Stress and Salivary Cortisol Levels in Patients with Plaque Psoriasis"

_jpm, 2021, doi:10.3390/jpm11111069_

Round 1

Reviewer 1 Report

Dear Authors,

The manuscript by Paolo Gisondi et al, on "Psychological stress and salivary cortisol levels in patients with plaque psoriasis" is a good effort although it lacks originality.

Furthermore:

  1. Table 1 is misleading: You exhibit your results on salivary cortisol in the last line of the table under a patient (n=126) and control (n=116) lane, which is totally wrong.
  2. According to the results of the same table, your control group comes from a completely different background. How did these patients come for cancer screening? Was it an urgency? Was it scheduled? Did they come for a different reason and meanwhile they discovered a suspicious mole?
  3. This is important because it can be crucial for their stress levels. I suppose, if you consider yourself as a patient with a possible skin cancer to be checked, you don't go to the doctor stress-free... Therefore a degree of bias should be concerned as far as your control group is chocen. This must be clearly explained in order to make this research valid.
  4. In line 108 correct "filthy" to "fifty"

Author Response

Psychological stress and salivary cortisol levels in patients with plaque psoriasis

 Reviewer 1 report:

Dear Authors,

The manuscript by Paolo Gisondi et al, on "Psychological stress and salivary cortisol levels in patients with plaque psoriasis" is a good effort although it lacks originality.

Furthermore:

Table 1 is misleading: You exhibit your results on salivary cortisol in the last line of the table under a patient (n=126) and control (n=116) lane, which is totally wrong.

AUTHORS’ REPLY: Thank you for your comment. We agree with the reviewer that salivary cortisol was measured in 27 patients with psoriasis and in 24 controls and we reported this measure only the text (results section page 4) and we have removed the last line of Table 1.

 According to the results of the same table, your control group comes from a completely different background. How did these patients come for cancer screening? Was it an urgency? Was it scheduled? Did they come for a different reason and meanwhile they discovered a suspicious mole?

This is important because it can be crucial for their stress levels. I suppose, if you consider yourself as a patient with a possible skin cancer to be checked, you don't go to the doctor stress-free... Therefore a degree of bias should be concerned as far as your control group is chosen. This must be clearly explained in order to make this research valid.

AUTHORS’ REPLY: We agree with the reviewer that including patients with known suspicious lesions could have represented a bias. However, patients included in the control group came for non-urgent, scheduled skin cancer screening visits which are reserved for patients without known suspicious cutaneous lesions or previous history of melanoma (these latter patients are followed-up in a dedicated clinic). Case and controls share the same source population and consecutively attended our clinic. It is likely that any patient who goes to the doctor, regardless of the motivation, usually has a certain degree of apprehension about his health. We have therefore included in the manuscript (methods) the following sentence detailing inclusion criteria: “Inclusion criteria for controls were age >18 years, hospital admission for routine melanoma screening, i.e. non-urgent scheduled dermatological visits for patients without suspicious cutaneous lesions or previous history of melanoma.” Moreover, we included this as a limitation of the study in the discussion section. In particular, the choice of patients who come to hospital for routine melanoma screening as a control group is a limitation of the study because such patients could be stressed for the motivation of the visit. However, this bias may have only diluted a difference that has been observed with patients with psoriasis.

 In line 108 correct "filthy" to "fifty"

AUTHORS’ REPLY: Thank you for your comment. We have corrected the word in the manuscript.

Reviewer 2 Report

The authors reported the results of an interesting observational study aimed at accessing psychological stress and salivary cortisol levels in psoriatic patients.

Even if psychological stress is considered as a trigger factor in psoriasis, limited data are currently available on salivary cortisol in psoriatic patients.

Interestingly, the authors reported the presence of significantly lower levels of salivary cortisol in psoriatic patients than in controls, despite higher stress levels of the former, suggesting a possible role of an alteration in this axis. These findings may better clarify the role of psychological stress as a common trigger in psoriasis.

Major limitations of the Study: (i) the small population size, and (ii)the lack of follow-up data. ( already stated by the authors)
The manuscript is well written. The introduction provides sufficient background, includes all relevant references, and well clarify the aim of the study.

Methods are adequately described, as well as the results section.

Author Response

Psychological stress and salivary cortisol levels in patients with plaque psoriasis

 Reviewer 2 report:

The authors reported the results of an interesting observational study aimed at accessing psychological stress and salivary cortisol levels in psoriatic patients.

Even if psychological stress is considered as a trigger factor in psoriasis, limited data are currently available on salivary cortisol in psoriatic patients.

Interestingly, the authors reported the presence of significantly lower levels of salivary cortisol in psoriatic patients than in controls, despite higher stress levels of the former, suggesting a possible role of an alteration in this axis. These findings may better clarify the role of psychological stress as a common trigger in psoriasis.

Major limitations of the Study: (i) the small population size, and (ii)the lack of follow-up data. ( already stated by the authors)

The manuscript is well written. The introduction provides sufficient background, includes all relevant references, and well clarify the aim of the study.

Methods are adequately described, as well as the results section.

AUTHORS’ REPLY: We thank very much the reviewer for the supportive comment.

Round 2

Reviewer 1 Report

Dear Authors,

I accept your manuscript in the present form.